# Association between Dietary Diversity and Sociopsychological Factors and the Onset of Dyslipidemia after the Great East Japan Earthquake: Fukushima Health Management Survey

**DOI:** 10.3390/ijerph192214636

**Published:** 2022-11-08

**Authors:** Fumikazu Hayashi, Tetsuya Ohira, Shiho Sato, Hironori Nakano, Kanako Okazaki, Masanori Nagao, Michio Shimabukuro, Akira Sakai, Junichiro James Kazama, Mitsuaki Hosoya, Atsushi Takahashi, Masaharu Maeda, Hirooki Yabe, Seiji Yasumura, Hitoshi Ohto, Kenji Kamiya

**Affiliations:** 1Radiation Medical Science Center for the Fukushima Health Management Survey, Fukushima Medical University, 1 Hikariga-oka, Fukushima-City 960-1295, Japan; 2Department of Epidemiology, Fukushima Medical University School of Medicine, 1 Hikariga-oka, Fukushima-City 960-1295, Japan; 3Department of Physical Therapy, Fukushima Medical University School of Medicine, 1 Hikariga-oka, Fukushima-City 960-1295, Japan; 4Department of Diabetes, Endocrinology and Metabolism, Fukushima Medical University School of Medicine, 1 Hikariga-oka, Fukushima-City 960-1295, Japan; 5Department of Radiation Life Sciences, Fukushima Medical University School of Medicine, 1 Hikariga-oka, Fukushima-City 960-1295, Japan; 6Department of Nephrology and Hypertension, Fukushima Medical University School of Medicine, 1 Hikariga-oka, Fukushima-City 960-1295, Japan; 7Department of Pediatrics, Fukushima Medical University School of Medicine, Fukushima, 1 Hikariga-oka, Fukushima-City 960-1295, Japan; 8Department of Gastroenterology, Fukushima Medical University School of Medicine, 1 Hikariga-oka, Fukushima-City 960-1295, Japan; 9Department of Disaster Psychiatry, Fukushima Medical University School of Medicine, 1 Hikariga-oka, Fukushima-City 960-1295, Japan; 10Department of Neuropsychiatry, Fukushima Medical University School of Medicine, 1 Hikariga-oka, Fukushima-City 960-1295, Japan; 11Department of Public Health, Fukushima Medical University School of Medicine, 1 Hikariga-oka, Fukushima-City 960-1295, Japan; 12Research Institute for Radiation Biology and Medicine, Hiroshima University, 1-2-3 Kasumi, Minami-ku, Hiroshima-City 734-8553, Japan

**Keywords:** dyslipidemia, sociopsychological factor, dietary diversity score, disaster victims, Fukushima nuclear accident, Great East Japan Earthquake

## Abstract

This study aimed to clarify the relationship between the onset of low-density lipoprotein hypercholesterolemia (hyper-LDLemia), high-density lipoprotein hypocholesterolemia (hypo-HDLemia), and hyper-triglyceridemia (hyper-TGemia) and lifestyle/socio-psychological factors among Fukushima evacuation area residents after the Great East Japan Earthquake. Participants included 11,274 non-hyper-LDLemia, 16,581 non-hypo-HDLemia, and 12,653 non-hyper-TGemia cases in the Fiscal Year (FY) 2011. In FY2011, these participants underwent a health checkup and responded to a mental health and lifestyle survey. The onset of each disease was followed through FY2017. The evacuation experience was positively associated with the risk of hyper-LDLemia, hypo-HDLemia, or hyper-TGemia. Conversely, the middle high dietary diversity score was negatively associated with the onset of hyper-TGemia. Moreover, low sleep satisfaction was positively associated with hypo-HDLemia and hyper-TGemia. The “almost never” exercise habit was positively associated with hypo-HDLemia. Current smoking and audible nuclear power plant explosions were positively associated with the risk of hyper-TGemia. Drinking habits exhibited a negative association with the onset of hyper-LDLemia, hypo-HDLemia, and hyper-TGemia. The results of this study indicate the need for continuous improvement in lifestyle, as well as efforts to eliminate the impact of disasters to prevent the onset of dyslipidemia among disaster evacuees.

## 1. Introduction

Dyslipidemia is characterized by low-density lipoprotein hypercholesterolemia (hyper-LDLemia), high-density lipoprotein hypocholesterolemia (hypo-HDLemia), or hyper-triglyceridemia (hyper-TGemia) [1]. Dyslipidemia was reported as an important risk factor for atherosclerosis, which can mimic myocardial infarction and stroke [2,3]. Conversely, a study using health checkup data from the Fukushima Health Management Survey, which was initiated after the Great East Japan Earthquake of 11 March 2011, reported a significant increase in the percentage of dyslipidemia among the residents of the area, from 49.8% in Fiscal Year (FY) 2011–2012 to 53.3% in FY2013–2014 [4]. Furthermore, the prevalence of hypo-HDLemia increased significantly after the earthquake, from 6.0% to 7.2%, and the evacuation experience was positively associated with the onset of hypo-HDLemia [5]. Additionally, a significantly greater number of new hyper-LDLemia cases were detected among evacuees vs. non-evacuees [6]. Therefore, residents in the evacuation area of Fukushima Prefecture are considered at a high risk of developing new dyslipidemia disorders, such as hyper-LDLemia, hypo-HDLemia, and hyper-TGemia, triggered by the impacts of the earthquake. These findings suggest that the examination of earthquake-related factors for dyslipidemia among residents of evacuation zones is an urgent issue when considering the prevention of cardiovascular diseases among residents of evacuation areas.

Additionally, it has recently become clear that lifestyle, such as eating habits, and sociopsychological factors, such as depressive states, are associated with an increased risk of dyslipidemia and metabolic syndrome, of which dyslipidemia is one component [7,8,9,10]. However, the relationship between socioeconomic and sociopsychological factors and the onset of dyslipidemia after the Great East Japan Earthquake has not been examined among residents of evacuated areas in Fukushima Prefecture. Therefore, this study aimed to clarify the impacts of lifestyle, sociopsychological, and earthquake-related factors on the onset of hyper-LDLemia, hypo-HDLemia, and hyper-TGemia after the Great East Japan Earthquake.

## 2. Materials and Methods

### 2.1. Participants

A flow chart of this study is provided in Figure 1. The Fukushima Health Management Survey was conducted among residents of the 13 municipalities that were evacuated after the Great East Japan Earthquake [11]. The participants of this study were 26,619 individuals aged between 40 and 90 years (11,382 men and 15,237 women; average age, 61.9 ± 11.0 years) who had undergone health checkups and responded to the questionnaire of the Mental Health and Lifestyle Survey (valid responses, 73,431; response rate, 40.7%) in FY2011. Among them, individuals with a history of or receiving medication for hyper-LDLemia, hypo-HDLemia, or hyper-TGemia in FY2011 were excluded. Baseline data for hyper-LDLemia were obtained from 13,590 individuals (6564 men and 7026 women), for hypo-HDLemia from 19,906 individuals (8460 men and 11,446 women), and for hyper-TGemia from 17,902 individuals (7503 men and 10,399 women). In addition, participants with an onset of each disease between 2012 and 2017 were followed up with, excluding those who were untraceable. Finally, risk factors for the occurrence of each disease were analyzed in 11,274 individuals (5423 men and 5851 women; mean age, 61.6 ± 11.3 years; mean follow-up, 3.6 ± 2.0 years) in the case of hyper-LDLemia, 16,581 individuals (6941 men and 9640 women; mean age, 61.3 ± 10.7 years; mean follow-up, 4.0 ± 1.9 years) in the case of hypo-HDLemia, and 12,653 individuals (5101 men and 7552 women; mean age, 61.4 ± 10.8 years; mean follow-up, 3.7 ± 1.9 years) in the case of hyper-TGemia.

In this study, disease states, body mass index (BMI), earthquake-related sociopsychological and socioeconomic factors; and lifestyle habits in FY2011 were examined regarding their impacts on the occurrence of hyper-LDLemia, hypo-HDLemia, or hyper-TGemia.

### 2.2. Definitions of Variables

#### 2.2.1. Lifestyle-Related Disease

Hyper-LDLemia was defined as an LDL cholesterol level > 140 mg/dL, or the use of cholesterol-lowering drugs. Hypo-HDLemia was defined as an HDL cholesterol level < 40 mg/dL, or the use of cholesterol-lowering drugs. Hyper-TGemia was defined as a fasting triglyceride level > 150 mg/dL, or the use of cholesterol-lowering drugs. Diabetes mellitus was defined as a fasting blood glucose level ≥ 126 mg/dL, casual blood glucose level ≥ 200 mg/dL, an HbA1c level ≥ 6.5% (National Glycohemoglobin Standardization Program [NGSP] criteria), or the use of hypoglycemic agents. Hepatic dysfunction was defined as an aspartate aminotransferase (AST) activity ≥ 31 U/L, an alanine aminotransferase (ALT) activity ≥ 31 U/L, or a γ-glutamyl transferase (γ-GT) activity ≥ 51 U/L. Renal dysfunction was defined as an estimated glomerular filtration rate (eGFR) < 60 mL/min/1.73 m^2^, a urinary protein > 1+, or being treated for chronic renal failure (including dialysis). The BMI was calculated using height and weight based on the following formula.

BMI (kg/m^2^) = weight (kg)/(height (m^2^). This study was based on the Japan Society for the Study of Obesity [12], which defines obesity as a BMI ≥ 25 kg/m^2^, with a lean status being defined as a BMI < 18.5 kg/m^2^.

#### 2.2.2. Dietary Diversity Score

The Food Frequency Questionnaire (FFQ) was used in this study to examine food intake [13]. The FFQ is a modified version of the questionnaire used in the Hiroshima and Nagasaki Life Span Study, the validity of which has been previously reported [13,14]. It has been reported in several national and international studies that the quality and pattern of diet, as well as specific nutrients and foods, affect dyslipidemia [15]. Therefore, in this study, we decided to evaluate the dietary diversity to assess the impact of food intake frequency on each disease. Although multiple methods exist to assess dietary diversity [16,17,18], appropriate assessment should be used due to differences in food culture among countries. For this reason, we referred to studies reported in Japan [16]. Therefore, rice and bread as cereals; fish as seafood; chicken, pork or beef, and ham or sausage as meat; green leafy vegetables, red or yellow vegetables, light-colored vegetables, and vegetable juice as vegetables; fruits and fruit juice as fruits; natto, miso soup, boiled beans, tofu and soy milk as beans products; and milk and yogurt as milk products were redefined as the seven food groups analyzed here. A score of 1 point was given if any of the foods in each food group was eaten daily, and 0 points were attributed otherwise; the total score was used as the dietary diversity score (DDS) (minimum, 0 points; maximum, 7 points) in this study. The DDS was further classified as low (0–1 points) for the first quartile or less, middle low (2 points) for the first to second quartiles, middle high (3 points) for the second to third quartiles, and high (4–7 points) for the third quartile or more, and their association with the occurrence of each disease was examined.

#### 2.2.3. Lifestyle and Sociopsychological Factor

The respondents were asked about their highest educational level using a four-question method: “elementary school, middle school;” “high school;” “vocational school, junior college;” and “university, graduate school.” Regarding their work situation after the earthquake, we assessed whether the respondents changed their jobs or lost their jobs. Rarely drinking (less than once a month), former drinking, current drinking (ethanol intake < 44.0 g [two glasses of sake] per day), and heavy drinking (ethanol intake ≥ 44.0 g per day) were defined on the basis of the questionnaire pertaining to drinking habits and the sake equivalent of drinks per day. The post-traumatic stress disorder checklist (PCL) [19] is a questionnaire that is used to measure trauma symptoms. On the basis of the results of previous studies [20,21], the presence of trauma symptoms was defined as having a total PCL score of ≥44. The evacuation experience, the experience of the Great East Japan Earthquake (tsunami, nuclear power plant accident [explosion heard]), smoking habits, sleep satisfaction, exercise habits, definition of the anxiety related to the impact of radiation on health due to radiation exposure with acute (e.g., death within 1 month), later-year (e.g., development of cancer), and next-generation (e.g., one’s children and grandchildren to be born in the future) nature, which was based on previous studies [22].

### 2.3. Statistical Methods

SAS 9.4 (SAS Institute Inc., Cary, NC, USA) was used for statistical analysis. Basic data were expressed as the mean ± standard deviation and number of persons (%). Unpaired t-tests or chi-squared tests were used to compare differences in each item according to the occurrence of each disease. Additionally, a sex-and-age-adjusted multivariate Cox regression analysis was used to identify the risk factors involved in the occurrence of each disease, and hazard ratio (HR) and 95% confidence interval (95% CI) values were obtained. The criterion for inclusion in the multivariate adjusted model was *p* < 0.100. BMI, Diabetes mellitus, liver dysfunction, and renal dysfunction were included in the Cox regression analysis model, to account for confounders from other diseases. Missing data were complemented with dummy variables. Statistical significance was set at *p* < 0.05 with a two-tailed test.

Statistical analysis was used for studying the correlations and confirming the multicollinearity among the factors. First, we evaluated the correlation using Spearman’s rank correlation coefficient. Factors that were correlated and entered into the multivariate Cox regression analysis were checked for variance inflation factor (VIF) using multiple regression analysis to confirm multicollinearity. It was found that VIF was <10 among all the factors used in the multivariate analysis, indicating no multicollinearity. Furthermore, previous studies reported that the evacuation experience was associated with many diseases. Therefore, we evaluated the interaction between the evacuation experience, diabetes, and other diseases; however, no association was found between evacuation experience and any of the studied diseases.

## 3. Results

### 3.1. Characteristics of the Participants

The physical factors of the participants, analyzed for each disease according to the no-incidence and incidence groups, are shown in Table 1. After 7 years of post-disaster observation, we found that 4102 of 11,274 people had developed hyper-LDLemia (incidence rate, 101.1), 4109 out of 16,581 people had developed hypo-HDLemia (incidence rate, 62.0), and 4217 out of 12,653 people had developed hyper-TGemia (incidence rate, 90.1) in FY2017.

### 3.2. Results of Hyper-LDLemia

The disaster-related and sociopsychological factors associated with hyper-LDLemia are shown in Table 2. The results of the chi-squared test revealed that sleep satisfaction, drinking habits, smoking habits, evacuation experience, job loss experience, trauma symptoms, anxiety about the impact of radiation on the next generation, and DDS were significantly associated with the occurrence of hyper-LDLemia.

Subsequently, a Cox regression analysis was performed using disaster-related and sociopsychological factors to calculate the HR for hyper-LDLemia (Table 3). In this analysis, variables that showed *p* < 0.100 were included in the final multivariate model.

A multivariate Cox regression analysis was performed to determine whether these factors are independently involved in the development of hyper-LDLemia (Table 4). The results showed that obesity (HR: 1.21, 95% confidence interval [95% CI]: 1.13–1.30), diabetes mellitus (HR: 1.21, 95% CI: 1.10–1.34), liver dysfunction (HR: 1.12, 95% CI: 1.04–1.20), renal dysfunction (HR: 1.17, 95% CI: 1.07–1.28), and evacuation experience (HR: 1.18, 95% CI: 1.10–1.26) were positively associated with the occurrence of hyper-LDLemia. Conversely, male sex (HR: 0.78, 95% CI: 0.72–0.85), lean body mass (HR: 0.69, 95% CI: 0.59–0.81), drinking habits (HR: 0.84, 95% CI: 0.78–0.90), heavy drinking (HR: 0.68, 95% CI: 0.60–0.77), and middle low DDS (HR: 0.88, 95% CI: 0.79–0.97) were negatively associated with the occurrence of hyper-LDLemia.

### 3.3. Results of Hypo-HDLemia

The disaster-related and sociopsychological factors related to hypo-HDLemia are shown in Table 2. The results of the chi-squared test revealed that education background, exercise habits, sleep satisfaction, drinking habits, smoking habits, experience of tsunami, experience of hearing nuclear power plant explosions, evacuation, job change, job loss, trauma symptoms, anxiety about acute impacts of radiation, anxiety about later-year impacts of radiation, anxiety about the impacts of radiation on the next generation, and DDS were significantly associated with the occurrence of hypo-HDLemia. A Cox regression analysis was then used to calculate the HR of risk factors for hypo-HDLemia (Table 3). In this analysis, variables that showed *p* < 0.100 were included in the final multivariate model. A multivariate Cox regression analysis was performed to identify factors that were independently involved in the development of hypo-HDLemia (Table 4). Age (HR: 1.02, 95% CI: 1.02–1.02), obesity (HR: 1.45, 95% CI: 1.36–1.55), diabetes mellitus (HR: 1.50, 95% CI: 1.37–1.64), liver dysfunction (HR: 1.20, 95% CI: 1.12–1.28), renal dysfunction (HR: 1.33, 95% CI: 1.23–1.44), rare exercise habits (HR: 1.11, 95% CI: 1.01–1.22), very dissatisfied sleep (HR: 1.22, 95% CI: 1.03–1.44), and evacuation experience (HR: 1.26, 95% CI: 1.17–1.35) were positively associated with the occurrence of hypo-HDLemia. Conversely, lean body mass (HR: 0.62, 95% CI: 0.52–0.75), drinking habit (HR: 0.77, 95% CI: 0.72–0.83), and heavy drinking (HR: 0.63, 95% CI: 0.55–0.71) were negatively associated with the occurrence of hypo-HDLemia.

### 3.4. Results of Hyper-TGemia

The disaster-related and sociopsychological factors related to hyper-TGemia are shown in Table 2. Educational background, level of sleep satisfaction, drinking habits, smoking habits, experience of tsunami, hearing the sound of nuclear power plant explosions, evacuation, job loss, trauma symptoms, and DDS were significantly associated with the occurrence of hyper-TGemia. Next, a Cox regression analysis was performed to calculate the HR of risk factors for hyper-TGemia (Table 3). In this analysis, variables that showed *p* < 0.100 were included in the final multivariate model. A multivariate Cox regression analysis was performed to identify factors that were independently involved in the development of hyper-TGemia (Table 4). AGE (HR: 1.01, 95% CI: 1.01–1.01), obesity (HR: 1.39, 95% CI: 1.31–1.49), diabetes mellitus (HR: 1.32, 95% CI: 1.20–1.46), liver dysfunction (HR: 1.30, 95% CI: 1.21–1.39), renal dysfunction (HR: 1.21, 95% CI: 1.11–1.31), quite unsatisfied sleep (HR: 1.12, 95% CI: 1.01–1.25), very dissatisfied sleep (HR: 1.23, 95% CI: 1.03–1.46), current smoking (HR: 1.20, 95% CI: 1.08–1.33), and evacuation (HR: 1.15, 95% CI: 1.07–1.23) were positively associated with the occurrence of hyper-TGemia. Whereas, lean body mass (HR: 0.50, 95% CI: 0.42–0.61), drinking habit (HR: 0.80, 95% CI: 0.75–0.86), and middle high DDS (HR: 0.90, 95% CI: 0.81–1.00) were negatively associated with the occurrence of hyper-TGemia.

## 4. Discussion

Previously, the authors of this study reported that an evacuation experience was positively associated with the occurrence of dyslipidemia after the Great East Japan Earthquake using health checkup data from the Fukushima Health Management Survey, with FY2011–2012 employed as the baseline and comparing it with data pertaining to FY2016–2017 [23]. Similarly, Satoh et al. reported that an evacuation experience was positively associated with the occurrence of hyper-LDLemia and hypo-HDLemia [5,6]. Consistent with the results of these studies, the present study showed that an evacuation experience was positively associated with the occurrence of hyper-LDLemia, hypo-HDLemia, and hyper-TGemia among the residents of the evacuation area. An evacuation experience has also been reported to be positively associated with obesity, diabetes mellitus, hypertension, metabolic syndrome, etc. [24]. The results of this study, which used data collected over 7 years after the earthquake, may indicate that various problems associated with the evacuation caused by the Great East Japan Earthquake have not been resolved, and still persist. Therefore, to prevent dyslipidemia, it is necessary to understand the problems caused by the Great East Japan Earthquake among those who experienced evacuation, and to identify ways to solve them.

A previous study presented an association between the living environment after the Great East Japan Earthquake, psychological distress, and disordered eating habits [13,25]. Additionally, evacuees had lower scores for healthy dietary patterns than non-evacuees, and in evacuees, these mean scores have been reported to decline over the years [26]. The present study showed that a middle-high or high DDS was negatively associated with, or tended to be negatively associated with, the development of hyper-TGemia. Conversely, in the case of hyper-LDLemia, the risk of developing this condition was reduced in individuals with a middle-low DDS, albeit with no statistically consistent trend, whereas no association was found in the case of hypo-HDLemia. Qorbani et al. performed a systematic review and meta-analysis, and reported that the association between the DDS and total cholesterol, LDL, and HDL levels was not significant, whereas its association with TG was significant [27]. A study of 160 patients with metabolic syndrome reported that those in the first quartile of the diversity score had higher serum triglyceride levels and systolic blood pressure, and lower serum adiponectin levels, than those in the fourth quartile [28]. These findings suggest that the residents of evacuation zones who exhibit reduced dietary diversity because of the disaster may be at an increased risk of developing hyper-TGemia. Therefore, we suggest that nutritional intervention to prevent a decrease in dietary diversity immediately after the disaster is necessary to prevent lipid abnormalities among the residents of evacuation zones.

This study showed that several of the psychological problems caused by the disaster, i.e., trauma symptoms, hearing the explosions from the nuclear accident, anxiety about the acute impacts of radiation, and low sleep satisfaction, which may reflect insomnia caused by psychological stress, were positively associated with, or tended to be associated with, the development of hyper-LDLemia, hypo-HDLemia, and hyper-TGemia. The psychological impacts of the Fukushima nuclear accident were extremely widespread, causing not only trauma symptoms but also chronic and more complex social problems, such as community and family fragmentation and stigma [29]. Leppold C, et al. have stated in their review that there is some evidence regarding the effect of exposure to multiple disasters on the patient’s mental health, physical health, and wellbeing, and that the potential risk of multiple disaster exposure exceeds that of single disaster exposure [30]. In Iwate Prefecture, which suffered damage from the Great East Japan Earthquake, and Miyagi Prefecture, surveys conducted soon after the disaster reported a high percentage of people with psychological disorders [31,32]. Sleep disturbances are a common health problem following traumatic events. Zhang W et al., conducted a cross-sectional study using data from the 2011 Fukushima Prefectural Health Survey and reported that 20.3% subjects were dissatisfied with their sleep [33]. Data from the 2013/2014 National Health and Nutrition Examination Survey (NHANES) that was con-ducted to evaluate the association between sleep and lipid profile, reported that short sleep duration was associated with low HDL-C and high triglycerides levels [34]. Additionally, in Japan, an analysis using data from the NHANES conducted for adults aged ≥20 years reported that sleep duration was associated with serum lipid and lipoprotein levels [35]. A study conducted 10 years after the Wenchuan earthquake reported that sleep disturbances still prevailed among the survivors, especially among men, and indicated that depression and PTSD are associated with the risk of sleep disturbances [36]. In addition, Tae H et al. reported that subjects with PTSD symptoms after the Sewol ferry disaster had lower serum HDL-C levels than those without PTSD [37]. These findings suggest that increasing stress and worsening psychological health conditions due to a disaster may reduce the duration of sleep, which may be a risk factor for dyslipidemia. Therefore, the regular monitoring of sleep status of may be important to prevent lipid abnormalities in the disaster victims. Furthermore, the experience of job loss tended to be positively associated with the development of hypo-HDLemia. Older adults and those with difficult living conditions reported stronger associations with a shift in psychological status toward deterioration, suggesting that worsening economic conditions increased psychological stress [38]. In Iwate Prefecture, which was affected by the earthquake, and Fukushima Prefecture, it was reported that the severe deterioration of economic conditions and lack of employment could be risk factors for poor mental health [31,39]. Furthermore, a study conducted ten years after the Wenchuan earthquake in China reported that some survivors still had PTSD symptoms and low income, chronic illness, and the death of an immediate family member due to the earthquake were associated with long-term PTSD symptoms [40]. A study conducted 25 years after the Spitak earthquake reported that predictors of PTSD symptom severity are destruction of the subject’s house, lack of treatment and social support, post-disaster adversity, and chronic medical illness [41]. The present study showed that PTSD symptoms are associated with the occurrence of hyper-LDLemia. These findings suggest that financial difficulties after a disaster may generate a vicious cycle of worsening psychological conditions, leading to an increase in the risk of chronic diseases such as dyslipidemia, which may lead to the further deterioration of psychological conditions. Therefore, it may be important to ensure that disaster victims do not experience financial difficulties through sustained financial support for preventing the occurrence of postdisaster dyslipidemia. In summary, the results of this study suggest that psychological trauma from the earthquake, exposure to radiation anxiety, and poor economic conditions due to job loss increase psychological stress and worsen the lipid profiles of the evacuation area residents. In addition, the increased risk of disease development due to disasters is not unique to the Great East Japan Earthquake, but one that is likely to occur in any disaster. Therefore, we need to be constantly prepared to provide psychological care and financial support in the early stages of a disaster.

The mechanisms via which psychological stress is associated with an increased risk of lipid abnormalities include increased cortisol secretion via the hypothalamus–pituitary–adrenal system [42]. Furthermore, psychological stress is thought to increase the expression of inflammatory cytokines and contribute to the development of hypertension, glucose intolerance, and dyslipidemia [43]. In recent years, it has become clear that psychological stress produces an inflammatory response [44]. In turn, several indices of inflammatory response have been noted to be associated with metabolic syndromes [45]. These findings suggest that the psychological stress caused by the earthquake induces inflammatory reactions, which may be a risk factor for the development of lifestyle-related diseases among the residents of the evacuation area. Therefore, care for psychological stress is very important for the prevention of lipid abnormalities among the residents of evacuation zones.

The present study showed that individuals who exercised little or only once per week were at a high risk, or tended to be at high risk, of developing hypo-HDLemia. Epidemiological studies have shown that hypo-HDLemia is a risk factor for vascular disease [46]. Therefore, if individuals continue to have a sedentary lifestyle, they are at a high risk of developing cardiovascular disease in the future. Conversely, the authors also reported that continued exercise and physical activity were important for preventing the development of dyslipidemia after the Great East Japan Earthquake [23]. Additionally, HDL has been reported to increase with regular exercise [47]. The authors suggest that the establishment of physical activity and exercise habits among the residents of evacuated areas is recommended in the future.

This study showed that drinking <44 g/day ethanol was negatively associated with the occurrence of hyper-LDLemia, hypo-HDLemia, and hyper-TGemia. Alcohol consumption is believed to have beneficial effects on blood lipids, including an increase in HDL cholesterol levels and a decrease in LDL cholesterol levels [48]. Conversely, alcohol consumption has been reported to be associated with increased triglyceride levels, with a J-curve relationship [49]. In the present study, drinking <44 g/day ethanol was also negatively associated with the occurrence of hyper-TGemia, whereas heavy drinking (ethanol intake ≥ 44 g/day) was not associated with this condition. Furthermore, the findings pertaining to the association between alcohol intake and LDL cholesterol have been reported to be inconsistent [50]. Thus, although alcohol consumption may work to improve lipid profiles, the type and amount of alcohol [51], genetic polymorphisms [48], and lifestyle [52] factors may affect this association. On the basis of these factors, the association between alcohol intake and lipid profile remains controversial. At the very least, heavy drinkers with poor lipid profiles may need to be advised to reduce their alcohol intake to improve their health.

In the present study, a current smoking habit was positively associated with the occurrence of hyper-TGemia. Smoking habits were reported to be associated with dyslipidemia [53]. Daily tobacco consumption has also been reported to be associated with lower HDL cholesterol levels and higher triglyceride levels [54], which is consistent with the results of this study. Therefore, it can be said that habitual smoking is a risk factor for dyslipidemia. Guidance regarding smoking may need to be further promoted to prevent dyslipidemia.

This study had several limitations. First, because only residents of evacuation areas were evaluated in this study, the results obtained may differ from the data generated when surveys are conducted in other areas. Therefore, future comparisons with areas that were not affected by the nuclear accident and subsequent evacuation may be necessary.

Second, the DDS in Kimura et al. [16] was calculated using ten food groups, excluding grains. However, the food diversity score in this study differs in that it queries seven food groups including grains, but excluding queries for eggs, seaweeds, and fat & oil, which are present in the original study. This study could not replicate the methodology of the original. Therefore, the validity of the questions and reproducibility of the results were not fully examined, thereby leaving room for further study. In the future, examining the association between dietary habits and dyslipidemia using different methods, such as dietary patterns, may be necessary to validate the present results.

Third, the age-adjusted prevalence of trauma symptoms is known to decrease annually, and studies have shown that the mental health of the residents of evacuated areas in Fukushima Prefecture has improved compared with that recorded at the time of the earthquake [55]. However, this study only assessed the impact of psychological stress as of FY2011, and could not take into account changes in psychological stress.

Fourth, Horikoshi et al. reported that the individuals who did not respond to the mental health survey had significantly higher rates of psychological distress than those who did [56]. However, it is unclear whether the respondents to the mental health and lifestyle survey reflect the psychosocial status of the entire population of the evacuation zones and other areas, because they represent only a portion of the population covered by the Fukushima Prefectural People’ s Health Survey.

## 5. Conclusions

In summary, this study reported that the impacts of disaster-related factors, such as evacuation and anxiety about radiation, continue to affect the onset of dyslipidemia among the residents of evacuated areas in Fukushima Prefecture in the long term, even seven years after the disaster. Furthermore, lifestyle factors such as dietary diversity, alcohol consumption, and smoking habits, were also implicated in the development of dyslipidemia. Therefore, to prevent the occurrence of dyslipidemia, it is necessary to provide continuous guidance on diet and smoking habits in cooperation with the local community, and to implement measures to eliminate the impacts of the disaster.

## Figures and Tables

**Figure 1 ijerph-19-14636-f001:**
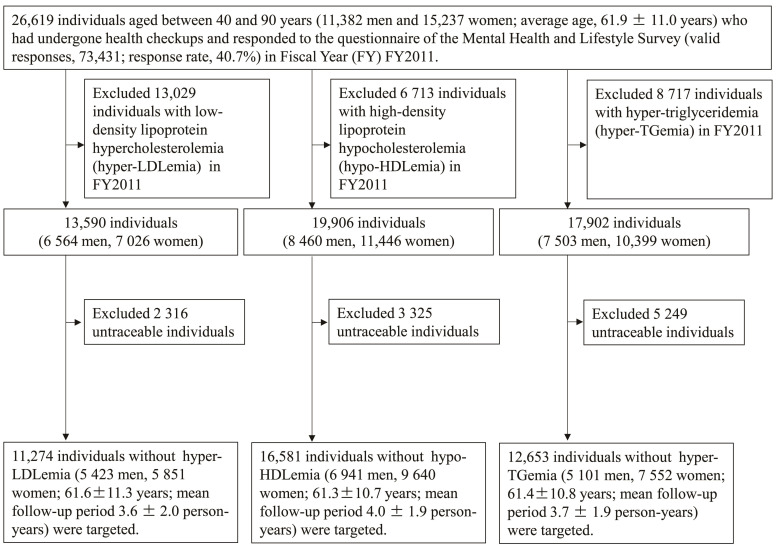
Flow chart of the selection of the research participants.

**Table 1 ijerph-19-14636-t001:** Comparison of the physical characteristics of the participants.

		Low-Density Lipoprotein Hypercholesterolemia			High-Density Lipoprotein Hypocholesterolemia			Hyper-Triglyceridemia		
Number of participants		11,274			16,581			12,653		
Follow-up period	Person/years	3.6 ± 2.0			4.0 ± 1.9			3.7 ± 1.9		
Incidence rate	1000 person/years	101.1			62.0			90.1		
		No incident	Incident case	*p* value	No incident	Incident case	*p* value	No incident	Incident case	*p* value
Number of individuals at risk		7172	4102		12,472	4109		8436	4217	
Age (years)		61.8 ± 11.7	61.2 ± 10.6		60.6 ± 11.0	63.3 ± 9.6		61.0 ± 11.3	62.2 ± 9.7	
Follow-up period	Person/years	4.3 ± 1.8	2.3 ± 1.6		4.5 ± 1.7	2.4 ± 1.6		4.4 ± 1.7	2.4 ± 1.6	
Sex	Men	3699 (51.6)	1724 (42.0)	<0.01	5161 (41.4)	1780 (43.3)	0.03	3318 (39.3)	1783 (42.3)	<0.01
	Women	3473 (48.4)	2378 (58.0)		7311 (58.6)	2329 (56.7)		5118 (60.7)	2434 (57.7)	
High-density lipoprotein cholesterol (mg/dL)		62 ± 17	59 ± 16		64 ± 14	57 ± 13		65 ± 15	59± 14	
Low-density lipoprotein cholesterol (mg/dL)		105 ± 20	120 ± 17		125 ± 30	145 ± 35		122 ± 28	142 ± 34	
Triglyceride (mg/dL)		100 ± 71	122 ± 95		100 ± 59	129 ± 75		79 ± 27	101 ± 28	
BMI (kg/m^2^)		23.2 ± 3.3	23.8 ± 3.4		23.2 ± 3.3	24.3 ± 3.3		22.9 ± 3.2	24.0 ± 3.2	
BMI	Standard (BMI ≤ 18.5 to <25.0 kg/m^2^)	4797 (66.9)	2604 (63.5)	<0.01	8419 (67.5)	2402 (58.5)	<0.01	5885 (69.8)	2645 (62.7)	<0.01
	Lean (BMI < 18.5 kg/m^2^)	449 (6.3)	169 (4.1)		697 (5.6)	111 (2.7)		566 (6.7)	113 (2.7)	
	Obesity (25.0 kg/m^2^ ≤ BMI)	1921 (26.8)	1329 (32.4)		3350 (26.9)	1594 (38.8)		1982 (23.5)	1458 (34.6)	
Fasting blood glucose (mg/dL)		100 ± 20	101 ± 20		99 ± 19	104 ± 23		99 ± 18	102 ± 21	
Casual blood glucose (mg/dL)		106 ± 38	111 ± 42		104 ± 34	114 ± 44		-	-	
Hemoglobin A1c (NGSP) (%)		5.4 ± 0.6	5.5 ± 0.7		5.4 ± 0.6	5.6 ± 0.8		5.4 ± 0.6	5.5 ± 0.7	
Medication for diabetes mellitus	No	6699 (94.6)	3761 (93.2)		11,807 (96.0)	3750 (92.5)		7973 (95.9)	3884 (93.2)	
	Yes	386 (5.4)	274 (6.8)		496 (4.0)	304 (7.5)		344 (4.1)	285 (6.8)	
Diabetes mellitus	No	6415 (90.5)	3583 (88.6)	<0.01	11,343 (92.1)	3489 (85.9)	<0.01	7692 (92.4)	3678 (88.1)	<0.01
	Yes	672 (9.5)	461 (11.4)		967 (7.9)	574 (14.1)		631 (7.6)	496 (11.9)	
Aspartate aminotransferase (U/L)		25 ± 15	25 ± 18		24 ± 13	26 ± 15		23 ± 9	25± 18	
Alanine aminotransaminase (U/L)		21 ± 17	23 ± 25		21 ± 16	24 ± 19		20 ± 13	23± 18	
γ-glutamyl transferase (U/L)		37 ± 60	36 ± 51		35 ± 47	39 ± 56		29 ± 33	36 ± 45	
Liver dysfunction	No	5156 (71.9)	2928 (71.4)	0.56	9192 (73.7)	2805 (68.3)	<0.01	6602 (78.3)	2950 (70.0)	<0.01
	Yes	2016 (28.1)	1174 (28.6)		3280 (26.3)	1304 (31.7)		1834 (21.7)	1267 (30.0)	
Estimated glomerular filtration rate (mL/min/1.73 m^2^)		73.8 ± 14.2	72.7 ± 13.8		74.0 ± 13.5	71.4 ± 13.7		74.0 ± 13.4	72.1 ± 13.4	
Proteinuria	Negative	7045 (98.5)	4035 (98.7)		12,309 (99.0)	4023 (98.1)		8334 (99.0)	4142 (98.4)	
	Positive	104 (1.5)	52 (1.3)		122 (1.0)	78 (1.9)		80 (1.0)	66 (1.6)	
Renal dysfunction	No	6062 (84.7)	3390 (82.9)	0.01	10,790 (86.8)	3274 (79.8)	<0.01	7302 (86.7)	3474 (82.6)	<0.01
	Yes	1093 (15.3)	699 (17.1)		1648 (13.2)	829 (20.2)		1116 (13.3)	734 (17.4)	

Data are presented as the number with a percentage or as the mean with standard deviation. Differences between the no incident and incident groups were analyzed using the chi-squared test. *p* < 0.05 were considered to indicate statistical significance. BMI, Body mass index; NGSP, National Glycohemoglobin Standardization Program.

**Table 2 ijerph-19-14636-t002:** Results of the analysis of earthquake-related, sociopsychological, and lifestyle factors of the participants.

		Low-Density Lipoprotein Hypercholesterolemia			High-Density Lipoprotein Hypocholesterolemia			Hyper-Triglyceridemia		
		No Incident	Incident Case	*p* Value	No Incident	Incident Case	*p* Value	No Incident	Incident Case	*p* Value
Number of individuals at risk		7172	4102		12,472	4109		8436	4217	
Dietary diversity score	Low (0–1)	1088 (20.5)	693 (22.1)	0.03	2136 (22.2)	618 (20.0)	<0.01	1326 (20.4)	702 (21.5)	0.628
	Middle low (2)	1576 (29.7)	838 (26.8)		2707 (28.2)	815 (26.3)		1801 (27.7)	892 (27.3)	
	Middle high (3)	1264 (23.8)	747 (23.9)		2275 (23.7)	749 (24.2)		1567 (24.1)	767 (23.5)	
	High (4–7)	1383 (26.0)	852 (27.2)		2498 (26.0)	912 (29.5)		1817 (27.9)	909 (27.8)	
Educational background	Elementary and junior high school	2003 (29.0)	1056 (26.7)	0.09	3102 (25.8)	1175 (29.7)	<0.01	2098 (25.9)	1103 (27.1)	<0.01
	High school	3456 (50.0)	2047 (51.7)		6178 (51.4)	2027 (51.3)		4144 (51.1)	2148 (52.8)	
	Vocational school/junior college	973 (14.1)	575 (14.5)		1916 (15.9)	514 (13.0)		1306 (16.1)	578 (14.2)	
	University/graduate school	484 (7.0)	279 (7.1)		820 (6.8)	238 (6.0)		566 (7.0)	242 (5.9)	
Exercise habit	Almost every day	1364 (19.5)	722 (18.0)	0.14	2176 (17.9)	729 (18.2)	<0.01	1510 (18.4)	753 (18.3)	0.17
	2–4 times per week	1716 (24.5)	953 (23.8)		2873 (23.6)	1069 (26.7)		2011 (24.5)	1061 (25.7)	
	Approximately once per week	1006 (14.4)	596 (14.9)		1754 (14.4)	625 (15.6)		1179 (14.3)	619 (15.0)	
	Almost never	2910 (41.6)	1730 (43.2)		5376 (44.1)	1583 (39.5)		3523 (42.8)	1688 (41.0)	
Level of sleep satisfaction	Satisfied	2154 (36.9)	1105 (32.7)	<0.01	3546 (34.7)	1086 (32.8)	<0.01	2394 (34.7)	1104 (32.4)	<0.01
	Slightly unsatisfied	2610 (44.7)	1563 (46.3)		4729 (46.3)	1521 (45.9)		3213 (46.6)	1585 (46.5)	
	Quite unsatisfied	857 (14.7)	547 (16.2)		1538 (15.1)	528 (15.9)		1026 (14.9)	548 (16.1)	
	Very dissatisfied	218 (3.7)	160 (4.7)		402 (3.9)	180 (5.4)		260 (3.8)	169 (5.0)	
Drinking habit	Never drinks or rarely drinks	2901 (41.3)	1982 (49.3)	<0.01	5780 (47.3)	2128 (52.8)	<0.01	4038 (48.8)	2141 (51.7)	<0.01
	Former drinking	213 (3.0)	118 (2.9)		286 (2.3)	126 (3.1)		207 (2.5)	118 (2.8)	
	Current drinking (ethanol intake < 44.0 g/day)	2921 (41.6)	1530 (38.1)		4827 (39.5)	1425 (35.4)		3330 (40.3)	1461 (35.2)	
	Heavy drinking (ethanol intake ≥ 44.0 g/day)	988 (14.1)	391 (9.7)		1338 (10.9)	348 (8.6)		697 (8.4)	425 (10.3)	
Smoking habit	Never smoked	3818 (54.8)	2438 (61.4)	<0.01	7455 (61.6)	2454 (61.6)	0.07	5318 (64.9)	2552 (62.2)	0.01
	Former smoking	1955 (28.1)	979 (24.7)		2861 (23.6)	993 (24.9)		1843 (22.5)	990 (24.1)	
	Current smoking	1190 (17.1)	554 (14.0)		1785 (14.8)	538 (13.5)		1029 (12.6)	562 (13.7)	
Experience of tsunami	No	5788 (80.7)	3276 (79.9)	0.28	10,098 (81.0)	3245 (79.0)	<0.01	6807 (80.7)	3327 (78.9)	0.02
	Yes	1384 (19.3)	826 (20.1)		2374 (19.0)	864 (21.0)		1629 (19.3)	890 (21.1)	
Experience of nuclear accident (explosion heard)	No	3295 (45.9)	1817 (44.3)	0.09	5836 (46.8)	1704 (41.5)	<0.01	3970 (47.1)	1761 (41.8)	<0.01
	Yes	3877 (54.1)	2285 (55.7)		6636 (53.2)	2405 (58.5)		4466 (52.9)	2456 (58.2)	
Experience of evacuation	No	3357 (47.0)	1601 (39.2)	<0.01	5738 (46.2)	1529 (37.4)	<0.01	3943 (47.0)	1649 (39.3)	<0.01
	Yes	3782 (53.0)	2480 (60.8)		6683 (53.8)	2557 (62.6)		4453 (53.0)	2551 (60.7)	
Job change	No	6952 (96.9)	3990 (97.3)	0.31	12,092 (97.0)	4016 (97.7)	<0.01	8199 (97.2)	4111 (97.5)	0.33
	Yes	220 (3.1)	112 (2.7)		380 (3.0)	93 (2.3)		237 (2.8)	106 (2.5)	
Job loss	No	5581 (77.8)	3058 (74.5)	<0.01	9547 (76.5)	3051 (74.3)	<0.01	6508 (77.1)	3166 (75.1)	<0.01
	Yes	1591 (22.2)	1044 (25.5)		2925 (23.5)	1058 (25.7)		1928 (22.9)	1051 (24.9)	
PTSD symptoms	No	5355 (79.4)	2948 (75.6)	<0.01	9376 (79.1)	2914 (75.3)	<0.01	6349 (79.4)	3065 (76.9)	<0.01
	Yes	1389 (20.6)	953 (24.4)		2470 (20.9)	954 (24.7)		1649 (20.6)	919 (23.1)	
Anxiety regarding the acute health impact of radiation	Extremely low to high	6288 (93.3)	3601 (93.1)	0.70	11,069 (94.0)	3566 (92.4)	<0.01	7484 (94.0)	3714 (93.2)	0.09
	Extremely high	454 (6.7)	268 (6.9)		709 (6.0)	295 (7.6)		480 (6.0)	272 (6.8)	
Anxiety regarding the impact of radiation on health in later years	Extremely low to high	5172 (76.2)	2907 (75.0)	0.15	9011 (76.2)	2881 (74.3)	0.02	6080 (76.1)	3013 (75.2)	0.25
	Extremely high	1615 (23.8)	971 (25.0)		2818 (23.8)	995 (25.7)		1905 (23.9)	994 (24.8)	
Anxiety regarding the impact of radiation on the health of the next generation	Extremely low to high	4379 (64.7)	2415 (62.4)	0.02	7603 (64.5)	2385 (61.7)	<0.01	5148 (64.6)	2491 (62.5)	0.02
	Extremely high	2390 (35.3)	1455 (37.6)		4187 (35.5)	1482 (38.3)		2824 (35.4)	1497 (37.5)	

Data are presented as the number with a percentage. The interval scale between the case and control groups was tested using the chi-squared test. PTSD, post-traumatic stress disorder. For two-sided testing, *p* < 0.05 was considered statistically significant.

**Table 3 ijerph-19-14636-t003:** Results of sex- and age-adjusted Cox regression analysis.

		Low-Density Lipoprotein Hypercholesterolemia		High-Density Lipoprotein Hypocholesterolemia		Hyper-Triglyceridemia	
Factor	Parameter	Sex- and Age-Adjusted HR ^a^ (95% CI) ^a^	*p* Value	Sex and Age-Adjusted HR (95% CI) ^a^	*p* Value	Sex and Age-Adjusted HR (95% CI) ^a^	*p* Value
Age	Continuous	1.00 (1.00–1.00)	0.31	1.02 (1.02–1.03)	<0.01	1.01 (1.01–1.01)	<0.01
Sex	Men (Ref. women)	0.74 (0.69–0.79)	<0.01	1.00 (0.94–1.07)	0.95	1.09 (1.02–1.16)	0.01
BMI	Lean (BMI < 18.5 kg/m^2^) (Ref. Standard (BMI ≤ 18.5 to <25.0 kg/m^2^))	0.69 (0.59–0.81)	<0.01	0.61 (0.51–0.74)	<0.01	0.50 (0.42–0.61)	<0.01
	Obesity (25.0 kg/m^2^ ≤ BMI)	1.28 (1.20–1.37)	<0.01	1.57 (1.48–1.67)	<0.01	1.50 (1.41–1.60)	<0.01
Diabetes mellitus	Yes (Ref. no)	1.30 (1.18–1.44)	<0.01	1.66 (1.52–1.82)	<0.01	1.47 (1.33–1.62)	0.01
Liver dysfunction	Yes (Ref. no)	1.15 (1.07–1.24)	<0.01	1.29 (1.21–1.39)	<0.01	1.42 (1.33–1.52)	<0.01
Renal dysfunction	Yes (Ref. no)	1.22 (1.12–1.33)	<0.01	1.40 (1.29–1.51)	<0.01	1.26 (1.16–1.37)	<0.01
Dietary diversity score	Middle low (2) (Ref. low (0–1))	0.85 (0.77–0.94)	<0.01	0.97 (0.87–1.07)	0.50	0.90 (0.82–1.00)	0.04
	Middle high (3)	0.90 (0.81–1.00)	0.05	0.99 (0.89–1.10)	0.84	0.86 (0.78–0.96)	<0.01
	High (4–7)	0.94 (0.84–1.04)	0.21	1.00 (0.90–1.11)	0.97	0.85 (0.77–0.94)	<0.01
Educational background	High school (Ref. elementary/junior high school)	1.04 (0.96–1.12)	0.38	0.99 (0.92–1.06)	0.71	1.00 (0.93–1.08)	0.93
	Vocational school/junior college	0.97 (0.87–1.09)	0.62	0.90 (0.80–1.00)	0.05	0.93 (0.83–1.03)	0.17
	University/graduate school	1.11 (0.97–1.27)	0.12	0.92 (0.80–1.06)	0.23	0.88 (0.76–1.01)	0.07
Exercise habit	2–4 times per week (Ref. almost every day)	1.00 (0.90–1.10)	0.94	1.11 (1.01–1.21)	0.04	1.04 (0.94–1.14)	0.46
	Approximately once per week	1.06 (0.95–1.18)	0.29	1.12 (1.01–1.25)	0.04	1.06 (0.96–1.18)	0.26
	Almost never	1.06 (0.96–1.16)	0.25	1.13 (1.03–1.24)	0.01	1.09 (0.99–1.19)	0.07
Level of sleep satisfaction	Slightly unsatisfied (Ref. satisfied)	1.07 (0.99–1.15)	0.11	1.08 (1.00–1.17)	0.06	1.09 (1.01–1.17)	0.04
	Quite unsatisfied	1.12 (1.01–1.24)	0.04	1.18 (1.06–1.31)	<0.01	1.16 (1.05–1.29)	<0.01
	Very dissatisfied	1.22 (1.03–1.44)	0.02	1.41 (1.20–1.65)	<0.01	1.33 (1.13–1.57)	<0.01
Drinking habit	Former drinking (Ref. never drinks or rarely drinks)	1.02 (0.84–1.23)	0.85	1.01 (0.84–1.21)	0.96	0.97 (0.81–1.18)	0.78
	Current drinking (ethanol intake < 44 g/day)	0.84 (0.78–0.90)	<0.01	0.77 (0.72–0.83)	<0.01	0.81 (0.75–0.87)	<0.01
	Heavy drinking (ethanol intake ≥ 44 g/day)	0.70 (0.62–0.79)	<0.01	0.66 (0.59–0.75)	<0.01	1.02 (0.9–1.14)	0.80
Smoking habit	Former smoking (Ref. never smoked)	1.01 (0.92–1.10)	0.88	1.03 (0.94–1.13)	0.52	1.08 (0.99–1.18)	0.10
	Current smoking	0.99 (0.89–1.10)	0.87	1.09 (0.98–1.21)	0.12	1.20 (1.08–1.33)	<0.01
Experience of tsunami	Yes (Ref. no)	1.09 (1.01–1.18)	0.02	1.08 (1.00–1.17)	0.04	1.08 (1.00–1.16)	0.05
Experience of nuclear accident (explosion heard)	Yes (Ref. no)	1.08 (1.01–1.15)	0.02	1.13 (1.07–1.21)	<0.01	1.13 (1.06–1.20)	<0.01
Experience of evacuation	Yes (Ref. no)	1.21 (1.14–1.29)	<0.01	1.32 (1.24–1.41)	<0.01	1.21 (1.13–1.28)	<0.01
Job change	Yes (Ref. no)	0.95 (0.79–1.15)	0.60	0.99 (0.80–1.21)	0.89	1.01 (0.83–1.23)	0.90
Job loss	Yes (Ref. no)	1.10 (1.02–1.18)	0.01	1.19 (1.10–1.27)	<0.01	1.10 (1.02–1.18)	0.01
PTSD symptoms	Yes (Ref. no)	1.16 (1.07–1.24)	<0.01	1.16 (1.08–1.25)	<0.01	1.10 (1.02–1.19)	<0.01
Anxiety regarding the acute health impact of radiation	Extremely high (Ref. extremely low to high)	1.07 (0.95–1.21)	0.28	1.23 (1.09–1.39)	<0.01	1.12 (0.99–1.27)	0.07
Anxiety regarding the impact of radiation on health in later years	Extremely high (Ref. extremely low to high)	1.06 (0.98–1.14)	0.14	1.10 (1.03–1.19)	<0.01	1.05 (0.98–1.13)	0.17
Anxiety regarding impact of radiation on the health of the next generation	Extremely high (Ref. extremely low to high)	1.09 (1.02–1.17)	<0.01	1.11 (1.04–1.18)	<0.01	1.09 (1.02–1.16)	0.01

^a^ Adjusted for age and sex. 95% CI, 95% confidence interval; HR, hazard ratio; Ref., reference; BMI, body mass index; PTSD, post-traumatic stress disorder. Cox proportional hazard model, *p* < 0.05 was considered statistically significant.

**Table 4 ijerph-19-14636-t004:** Results of the multivariate Cox regression analysis.

		Low-Density Lipoprotein Hypercholesterolemia		High-Density Lipoprotein Hypocholesterolemia		Hyper-Triglyceridemia	
Factor	Parameter	Adjusted HR (95% CI) ^a^	*p* Value	Adjusted HR (95% CI) ^b^	*p* Value	Adjusted HR (95% CI) ^c^	*p* Value
Age	Continuous	1.00 (0.99–1.00)	0.18	1.02 (1.02–1.02)	<0.01	1.01 (1.01–1.01)	<0.01
Sex	Men (Ref. women)	0.78 (0.72–0.85)	<0.01	1.06 (0.98–1.14)	0.16	0.97 (0.88–1.05)	0.43
BMI	Lean (BMI < 18.5 kg/m^2^) (Ref. Standard (BMI ≤ 18.5 to <25.0 kg/m^2^))	0.69 (0.59–0.81)	<0.01	0.62 (0.52–0.75)	<0.01	0.50 (0.42–0.61)	<0.01
	Obesity (25.0 kg/m^2^ ≤ BMI)	1.21 (1.13–1.30)	<0.01	1.45 (1.36–1.55)	<0.01	1.39 (1.31–1.49)	<0.01
Diabetes mellitus	Yes (Ref. no)	1.21 (1.10–1.34)	<0.01	1.50 (1.37–1.64)	<0.01	1.32 (1.20–1.46)	<0.01
Liver dysfunction	Yes (Ref. no)	1.12 (1.04–1.20)	<0.01	1.20 (1.12–1.28)	<0.01	1.30 (1.21–1.39)	<0.01
Renal dysfunction	Yes (Ref. no)	1.17 (1.07–1.28)	<0.01	1.33 (1.23–1.44)	<0.01	1.21 (1.11–1.31)	<0.01
Dietary diversity score	Middle low (2) (Ref. low (0–1))	0.88 (0.79–0.97)	0.01			0.93 (0.84–1.02)	0.13
	Middle high (3)	0.93 (0.84–1.04)	0.19			0.90 (0.81–1.00)	0.05
	High (4–7)	0.98 (0.88–1.09)	0.71			0.91 (0.82–1.00)	0.06
Educational background	High school (Ref. elementary/junior high school)			1.00 (0.92–1.07)	0.90	1.01 (0.93–1.09)	0.88
	Vocational school/junior college			0.92 (0.82–1.03)	0.14	0.97 (0.87–1.07)	0.51
	University/graduate school			0.94 (0.82–1.09)	0.41	0.93 (0.81–1.08)	0.34
Exercise habit	2–4 times per week (Ref. almost every day)			1.06 (0.97–1.17)	0.20	0.99 (0.90–1.09)	0.87
	Approximately once per week			1.11 (0.99–1.23)	0.07	1.03 (0.93–1.15)	0.55
	Almost never			1.11 (1.01–1.22)	0.03	1.03 (0.94–1.13)	0.52
Level of sleep satisfaction	Slightly unsatisfied (Ref. satisfied)	1.03 (0.95–1.11)	0.53	1.03 (0.95–1.12)	0.43	1.07 (0.98–1.15)	0.12
	Quite unsatisfied	1.03 (0.93–1.15)	0.55	1.09 (0.97–1.21)	0.14	1.12 (1.01–1.25)	0.04
	Very dissatisfied	1.08 (0.91–1.28)	0.41	1.22 (1.03–1.44)	0.02	1.23 (1.03–1.46)	0.02
drinking habit	Former drinking (Ref. never drinks or rarely drinks)	1.00 (0.82–1.21)	0.96	0.98 (0.82–1.18)	0.87	0.96 (0.79–1.16)	0.66
	Current drinking (ethanol intake < 44 g/day)	0.84 (0.78–0.90)	<0.01	0.77 (0.72–0.83)	<0.01	0.80 (0.75–0.86)	<0.01
	Heavy drinking (ethanol intake ≥ 44 g/day)	0.68 (0.60–0.77)	<0.01	0.63 (0.55–0.71)	<0.01	0.94 (0.83–1.06)	0.28
Smoking habit	Former smoking (Ref. never smoked)					1.07 (0.98–1.17)	0.16
	Current smoking					1.20 (1.08–1.33)	<0.01
Experience of tsunami	Yes (Ref. no)	1.06 (0.98–1.14)	0.17	1.03 (0.95–1.11)	0.48	1.03 (0.95–1.11)	0.49
Experience of nuclear accident (explosion heard)	Yes (Ref. no)	1.01 (0.94–1.07)	0.88	1.04 (0.98–1.12)	0.20	1.06 (0.99–1.13)	0.09
Experience of evacuation	Yes (Ref. no)	1.18 (1.10–1.26)	<0.01	1.26 (1.17–1.35)	<0.01	1.15 (1.07–1.23)	<0.01
Job loss	Yes (Ref. no)	1.01 (0.94–1.09)	0.77	1.07 (1.00–1.16)	0.06	1.01 (0.94–1.09)	0.72
PTSD symptoms	Yes (Ref. no)	1.08 (1.00–1.18)	0.05	1.04 (0.95–1.12)	0.41	0.99 (0.91–1.08)	0.85
Anxiety regarding the acute health impact of radiation	Extremely high (Ref. extremely low to high)			1.13 (0.99–1.29)	0.08	1.03 (0.90–1.17)	0.70
Anxiety regarding the impact of radiation on health in later years	Extremely high (Ref. extremely low to high)			0.99 (0.90–1.09)	0.82		
Anxiety regarding impact of radiation on the health of the next generation	Extremely high (Ref. extremely low to high)	1.05 (0.98–1.12)	0.18	1.04 (0.95–1.13)	0.42	1.03 (0.96–1.11)	0.39

^a^ Adjusted for age, sex, BMI, diabetes mellitus, liver dysfunction, renal dysfunction, dietary diversity score, level of sleep satisfaction, drinking habit, experience of tsunami, experience of nuclear accident (explosion heard), experience of evacuation, job loss, PTSD symptoms, and anxiety regarding the impact of radiation on the health of the next generation. ^b^ Adjusted for age, sex, BMI, diabetes mellitus, liver dysfunction, renal dysfunction, educational background, exercise habit, level of sleep satisfaction, drinking habit, experience of tsunami, experience of nuclear accident (explosion heard), experience of evacuation, job loss, PTSD symptoms, anxiety regarding the acute health impact of radiation, anxiety regarding the impact of radiation on health in later years, and anxiety regarding impact of radiation on the health of the next generation. ^c^ Adjusted for age, sex, BMI, diabetes mellitus, liver dysfunction, renal dysfunction, dietary diversity score, educational background, exercise habit, level of sleep satisfaction, drinking habit, smoking habit, experience of tsunami, experience of nuclear accident (explosion heard), experience of evacuation, job loss, PTSD symptoms, anxiety regarding the acute health impact of radiation, and anxiety regarding the impact of radiation on the health of the next generation. 95% CI, 95% confidence interval; HR, hazard ratio; Ref., reference; BMI, body mass index; PTSD, post-traumatic stress disorder. Cox proportional hazard model, *p* < 0.05 was considered statistically significant.

## Data Availability

The datasets analyzed during the present study are not publicly available because the data from the Fukushima Health Management Survey belong to the Fukushima Prefecture government and can only be used within that organization.

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
