# Peer review of "Association between Dietary Diversity and Sociopsychological Factors and the Onset of Dyslipidemia after the Great East Japan Earthquake: Fukushima Health Management Survey"

_ijerph, 2022, doi:10.3390/ijerph192214636_

Round 1

Reviewer 1 Report

   This is an interesting study analysing a health survey and
   revealing several factors estimating risk for dyslipidemia
   disorders.

   When I understand the methods correctly, the research participants
   were collected 2011 (Figure 1), and participants with on onset of
   dyslipidemia disorder between 2012 and 2017 were re-evaluated
   in 2017. Then risk factors were estimated using chi^2 tests and Cox
   regression models.

   Table 3 shows that many variables like Diabetes mellitus, Liver
   dysfunction, Renal dysfunction, Experience of evacuation etc. had
   an impact on occurrence of hyper-LDLemia. Do you have checked that variables like Diabetes mellitus, Liver dysfunction, Renal
   dysfunction, BMI, Experience of evacuation are not associated with
   each other? E.g. when 'renal dysfunction' and 'Experience of
   evacuation' are correlated, then this can bias interpretation of
   effects.

   My second question is related to this problem: Maybe some of the
   effects can be explained by underlying interactions of predictors,
   e.g. an interaction of "renal dysfunction' and 'Experience of
   evacuation?

Reviewer 2 Report

In this paper, the authors investigated the relationship between dietary diversity and sociopsychological factors after the great east Japan earthquake near Fukushima. This topic is attractive, especially from a sociopsychological view. Also, this research could benefit different fields, and the experimental data and results were valuable.

The main recommendation from the reviewer is that the authors can add more academic theories based on sociopsychology and discuss their results deeper from the sociopsychological view.

In addition, the reviewer thinks that the authors could also add some research on a different area for this earthquake, maybe Iwate or Ibaraki since the Fukushima nuclear power plant accident had a huge impact on the people who live in the Fukushima area. Furthermore, it is also interesting to discuss other post-disasters to compare their results in different countries and cultures. This might improve the versatility of their findings and show the advantages and limitations of this research.
